# Optimizing the Production of gp145, an HIV-1 Envelope Glycoprotein Vaccine Candidate and Its Encapsulation in Guanosine Microparticles

**DOI:** 10.3390/vaccines11050975

**Published:** 2023-05-12

**Authors:** Pearl Akamine, José A. González-Feliciano, Ruth Almodóvar, Gloriner Morell, Javier Rivera, Coral M. Capó-Vélez, Manuel Delgado-Vélez, Luis Prieto-Costas, Bismark Madera, Daniel Eichinger, Ignacio Pino, José H. Rivera, José Ortiz-Ubarri, José M. Rivera, Abel Baerga-Ortiz, José A. Lasalde-Dominicci

**Affiliations:** 1Clinical Bioreagent Center, Molecular Sciences Research Center, University of Puerto Rico, San Juan 00926, Puerto Ricomanuel.delgadovelez@upr.edu (M.D.-V.); abel.baerga@upr.edu (A.B.-O.); 2CDI Laboratories, Mayagüez 00680, Puerto Rico; 3Department of Biology, Río Piedras Campus, University of Puerto Rico, San Juan 00931, Puerto Rico; 4Department of Chemistry, Río Piedras Campus, University of Puerto Rico, San Juan 00925, Puerto Rico; 5Department of Computer Sciences, Río Piedras Campus, University of Puerto Rico, San Juan 00925, Puerto Rico; 6Department of Biochemistry, Medical Sciences Campus, University of Puerto Rico, San Juan 00936, Puerto Rico; 7Institute of Neurobiology, Medical Sciences Campus, University of Puerto Rico, San Juan 00901, Puerto Rico

**Keywords:** human immunodeficiency virus, envelope protein, gp145, vaccine, glycosylation, encapsulation, microparticles, supramolecular hacky sacks, manufacturing, scale-up

## Abstract

We have developed a pipeline to express, purify, and characterize HIV envelope protein (Env) gp145 from Chinese hamster ovary cells, to accelerate the production of a promising vaccine candidate. First in shake flasks, then in bioreactors, we optimized the growth conditions. By adjusting the pH to 6.8, we increased expression levels to 101 mg/L in a 50 L bioreactor, nearly twice the previously reported titer value. A battery of analytical methods was developed in accordance with current good manufacturing practices to ensure a quality biopharmaceutical. Imaged capillary isoelectric focusing verified proper glycosylation of gp145; dynamic light scattering confirmed the trimeric arrangement; and bio-layer interferometry and circular dichroism analysis demonstrated native-like properties (i.e., antibody binding and secondary structure). MALDI-TOF mass spectrometry was used as a multi-attribute platform for accurate mass determination, glycans analysis, and protein identification. Our robust analysis demonstrates that our gp145 product is very similar to a reference standard and emphasizes the importance of accurate characterization of a highly heterogeneous immunogen for the development of an effective vaccine. Finally, we present a novel guanosine microparticle with gp145 encapsulated and displayed on its surface. The unique properties of our gp145 microparticle make it amenable to use in future preclinical and clinical trials.

## 1. Introduction

Despite tremendous research efforts and prevention campaigns, the human immunodeficiency virus (HIV) remains a costly global health threat. As of 2020, 38 million individuals were living with HIV with 1.5 million new infections, 160,000 of them children under the age of fourteen years. Although the number of new infections has decreased by 25% since 2010, mostly thanks to educational campaigns together with the increased availability of anti-retroviral therapeutics, it has been stipulated that lowering the incidence of HIV globally will require the development and deployment of a vaccine [1,2].

The surface envelope glycoprotein (Env) of HIV has been the primary target for vaccine development during the last decades [3,4,5,6]. It is made by the virus as a single polypeptide, gp160, which is cleaved into gp120 and gp41 to form the mature protein. The membrane-spanning gp41 associates with the soluble gp120 to form a hetero-complex, which further self-associates to form an Env trimer [7]. Initial attempts at making a protein-based vaccine against HIV targeted the gp120 monomer, and this was eventually tested in the Thailand RV144 clinical trial [3]. Although the monomeric gp120 did not provide population-wide protection against the virus, it did reveal some serological correlates of protection or surrogate markers that have been useful in predicting the effectiveness of new vaccine candidates as they emerge, thus streamlining the process of evaluating vaccine efficacy [8,9].

The RV144 trial was followed by an enthusiastic drive to redesign the Env into constructs that preserve important immunological structural features of the protein, such as its trimeric quaternary structure and the native glycosylation profile that can drive the antibody response to specific epitopes. These newer and longer versions of the Env include: (1) uncleaved gp140, which consists of gp120 with additional portions of gp41 stopping just short of the membrane-proximal external region (MPER) [10]; (2) uncleaved gp145, which is similar to the gp140 but includes the destabilizing, immunogenic MPER region [11]; (3) native flexible linker (NFL) or uncleaved fusion optimized (UFO) gp140 in which the gp120 and gp41 portions are separated by an NFL [12,13]; and (4) SOSIP gp140, which contains gp120 and gp41 portions resulting from cleavage by the furin protease, where the cleavage products remain joined by an engineered disulfide bond [14,15].

An important obstacle to the large-scale production of these longer versions of the Env glycoprotein is manufacturability. For instance, the soluble gp120 has been known to give higher yields [10] and the longer Env construct which contains portions of the transmembrane gp41 is known to be problematic due to giving lower yields [16]. The production titers of recombinant Env are generally low. It has been estimated that the titers for Env glycoproteins are about 100-fold lower (~10 mg/L) than the titers for the production of monoclonal antibodies using mammalian cells [17]. There are numerous explanations for the low Env titers, ranging from abundant glycosylation to charge heterogeneity, both of which may affect protein secretion and product potency [15,18]. With an average of 27 glycosylation sites per molecule, depending on the strain, HIV Env is probably the most abundantly glycosylated therapeutic protein to be manufactured, and glycosylation is a bottleneck in production [19]. For instance, in the case of the SOSIP construct, there is an additional furin cleavage step that poses a potential speed bump toward manufacturability [20]. The sudden advent of nanoparticles to the therapeutic field was undoubtedly accelerated by the COVID-19 pandemic. Its use as an mRNA transporter demonstrated its feasibility for safe use in humans. Indeed, the use of these particles is not limited to encapsulating nucleic acids, but they can also be used to encapsulate and deliver immunogenic proteins [21], such as gp145. These obstacles can be overcome with careful optimization of the upstream process, together with the development of high-end protein analytics to ensure close monitoring across the entire manufacturing workflow [22,23,24]. To address these challenges, we established the UPR Clinical Bioreagent Center (UPR-CBC) [25], as invited by the White House HIV Policy Office and funded by the Vaccine Research at NIAID/NIH. The UPR-CBC is made up of the University of Puerto Rico, CDI Laboratories, Inc., and regional biotechnology firms that serve as manufacturing advisors. This effort aimed to optimize the production and manufacturing of the HIV Env glycoprotein immunogen gp145 for clinical trials. Our project at UPR-CBC was defined with a clear emphasis on progressing to manufacturing.

Here, we describe the UPR-CBC efforts to increase the titer and yield of the CO6980v0c22 uncleaved trimeric Env, hereafter called gp145 [11]. We were able to drive the titer from 40 mg/L to 160 mg/L by carefully selecting upstream parameters. We also describe a suite of analytical tests [25] aimed at ensuring that the quality attributes for this glycoprotein are not lost as we push for higher titers. All of our procedures are easily transferable to a cGMP-regulated manufacturing facility. Finally, for the first time, we present a patented guanosine microparticle which encapsulates and presents our antigen, gp145, and is ready for preclinical trials.

## 2. Materials and Methods

### 2.1. CHO-K1 Cell Line Expressing gp145

A Chinese hamster ovary (CHO)-K1 cell line previously selected for the stable expression of uncleaved gp145 CO6980v0c22 [11] was transferred from the Division of Acquired Immunodeficiency Syndrome (DAIDS), National Institute of Allergy and Infectious Diseases (NIAID), National Institutes of Health (material transfer agreement-MTA #980611). Briefly, the chemically-defined PowerCHO-2 conditioned media, supplemented with glutamine (4 mM) and puromycin (5 µg/mL), was inoculated with ≥4.6 × 10^5^ cells/mL in a 5 L bioreactor (4 L working volume). Cells were cultured at 37 °C, pH 7.2, and dO_2_ 50%, with 120 rpm agitation. Cells were harvested on day 7 and provided a maximum titer of 40 mg/L.

### 2.2. Upstream and Downstream Strategies to Increase gp145 Titers

Cell culture. Generally, seeding train, supplement studies, and media screening were performed in shake flasks. Initial optimization of growth conditions was performed in 250 mL shake flasks. For these studies, CHO-K1 cells were grown in PowerCHO-2 serum-free media supplemented with 1X penicillin/streptomycin, GlutaMAX (4 mM), and puromycin (5 µg/mL) at pH 7.2. The entire manufacturing process (upstream and downstream), as well as all analytical tests used to investigate gp145’s quality attributes, resulted in the development of auditable standard operating procedures (SOPs). The reagents and supplements used in this study are of the highest quality available, with the majority being USP grade. All instruments used in this study were validated and certified by their respective manufacturers and many meet 21 CFR Part 11 requirements.

Optimal pH determination. The pH of the cultures was optimized using the DASGIP parallel bioreactor system (Eppendorf). Briefly, each of the three small bioreactors was filled with 200 mL of PowerCHO-2 serum-free media (supplemented as described above). The pH of the media was adjusted by sparging with CO_2_ at 0.6 SLPM prior to inoculation with 0.5 × 10^6^ cells/mL. The cells were harvested after 9 to 11 days of growth (n = 2). Three pH set points were tested (pH 7.2, 7.0, and 6.8) and maintained with CO_2_. The dO_2_ set point was 35%, air and oxygen flow were 1 SLPH, and the agitation rate was 90 rpm. A control shake flask (SF) was run in parallel in a 5% CO_2_ incubator, with the same medium components, without pH adjustment.

Scale-up. The production of gp145 was scaled up in a Finesse G3 Lab Bioprocess Controller (Thermo Scientific) using BioBLU 50 L single-use bioreactors (40 L working volume, Eppendorf). CHO-K1 cells were grown for 11 days in PowerCHO-2 serum-free media (supplemented as described above with puromycin (10 µg/mL)) at pH 6.8. Prior to inoculation, the pH of the media was adjusted by sparging with CO_2_ at 0.6 SLPM. The agitation rate was set to 60 rpm and the air sparger at 1.5 SLPM. The dO_2_ set point was 35% and antifoam was used at 0.01%. The seeding density was 0.8 × 10^6^ cells/mL from passage 17, based on our studies that showed a seeding density of 0.8 × 10^6^ cells/mL produced higher titers (108 mg/L) than a seeding density of 0.5 × 10^6^ cells/mL (86 mg/L). The temperature was maintained using a heating blanket at 37 °C. The daily samples were not clarified. Viable cell density and the percent viability were determined using a Cellometer Auto T4 cell counter (Nexcelom) by staining cells with trypan blue. The growth performance was further monitored using a Cedex Bio Analyzer (Roche) for the following metabolites: glucose, glutamate, glutamine, GlutaMAX, ammonia (NH_3_), lactate, and lactate dehydrogenase (LDH).

Downstream. Harvested supernatants were subsequently clarified with SartoClear (10 g/L), sterile filtered, and then stored at −80 °C in 1 L aliquots. After thawing, the supernatant was concentrated using 100 kDa MWCO PES Vivaflow 200 System (Sartorius), then buffer-exchanged into the binding buffer (25 mM Tris pH 8.0, 1 mM CaCl_2_, and 1 mM MnCl_2_). Salts were added just before loading to the lentil lectin column (5 mL bed volume, Cytiva). The flow-through was collected and reloaded. The eluate was concentrated and exchanged into the TOYOPEARL binding buffer (25 mM Tris, pH 8.0). Samples were loaded onto a 10 mL TOYOPEARL NH_2_-750F column and eluted using a salt gradient elution from 0 to 1 M NaCl in 25 mM Tris, pH 8.0. The purified gp145 was buffer exchanged into 1X PBS. The purification was performed using the ÄKTA Avant 25 System (GE). The final purified gp145 was stored at −80 °C in sterile conical glass vials to prevent the glycoprotein loss that occurs in plastic tubes.

Feed strategies. Different feed strategies were explored using 1 L glass Finesse bioreactors to further optimize the production of gp145. Three feeding strategies were tested: first, two feeds were given on day three and day six; second, a condition to maintain glucose levels between 2 g/L and 3 g/L with a feed supplemented with 50 mM GlutaMAX. For the last strategy, glucose and GlutaMAX were both monitored daily and fed as necessary to maintain glucose levels between 4 and 6 g/L; third, maintaining glucose levels between 4 g/L and 6 g/L while replenishing GlutaMAX to 4 mM as needed. A control batch reactor was included without subjecting to any feed strategy. All reactors were seeded with 0.8 × 10^6^ cells/mL in PowerCHO-2 serum-free medium supplemented with 1X penicillin/streptomycin, GlutaMAX (4 mM), and puromycin (10 µg/mL). The operational parameters were as follows: pH 6.8, 37 °C, dO_2_ 35%, air and oxygen flow 0.03 SLPM, and agitation rate 90 rpm.

### 2.3. Titer Determination and Hydrophobic Footprint Analysis Using Reversed-Phase High-Performance Liquid Chromatography (RP-HPLC)

Titers were determined as described before [26]. The titer value is the average of three technical replicates and the error reported is the standard deviation. The hydrophobic footprint analysis was carried out using the same method as for titer determination with a minor change, namely (time/%B): 0/30, 2/30, 3.5/65, 4/70, 5/75, 6/75, 6.5/95, 9.5/95, and 10/30. To dilute gp145 supernatant samples, 1X Dulbecco’s phosphate buffered saline (DPBS) (Corning) was used and standards were prepared using spent media. All standards and samples were run in triplicate. The tail factor and retention time errors reported are the standard deviations.

### 2.4. Host Cell Protein (HCP) Determination and Binding Affinity Analysis Using Bio-Layer Interferometry

HCP quantification and gp145 binding studies were performed using the Octet^®^ QK^e^ (FortéBio) instrument. For HCP concentration determination, the anti-CHO host cell proteins (HCP) detection kit (FortéBio) was used, which utilizes the 3G anti-CHO HCP antibodies from Cygnus. The CHO HCP standard curve concentrations include 200, 100, 50, 25, 12.5, 6.25, and 3.13 ng/mL. The HCP values were determined at 4 different dilutions (1:5000, 1:10,000, 1:15,000, and 1:20,000) to ensure linearity of the readings. Each dilution was read in triplicate. The reported values are an average of 12 readings. To challenge the HCP quantitation method (detection sensitivity = 0.5 ng/mL, per manufacturer), the Day 4 sample was spiked with HCP (80 ng/mL), which resulted in a recovery of 98%. The K_D_ values for the gp145 protein were determined by loading the ligand, the 4E10 (3 µg/mL) broadly neutralizing antibody (Polymun Scientific), onto an anti-human IgG Fc capture (AHC) sensor (FortéBio, cat no. 18-5060). The serial dilutions of gp145 were used from 1–0.125 µM for curve fitting. The flow rate was 1200 rpm at 30 °C. All buffers contained 0.5X kinetics buffer (cat no. 18-5032, FortéBio). Data were analyzed using the FortéBio Analysis 8.2 program to obtain the rate of association (k_a_), rate of dissociation (k_d_), and dissociation constant (KD) [27]. A reference sensor was used to correct for background drift. 

### 2.5. Sodium Dodecyl Sulfate-Polyacrylamide Gel Electrophoresis (SDS-PAGE)

The SDS-PAGE Mini-PROTEAN TGX precast 10% gel (Bio-Rad) was run under reducing conditions as recommended by the manufacturer. The gel was run for 40 min at 200 V in 1X Tris/Glycine/SDS running buffer (Bio-Rad) and stained with Coomassie Blue. A 4–15% gradient gel (Bio-Rad) was run for 30 min at 50 V and then 2.5 h at 100 V at room temperature in 1X Tris/Glycine buffer for the native gel.

### 2.6. MALDI-TOF Mass Spectrometry

Whole protein analysis. The gp145 whole protein was analyzed in linear positive mode on an AB Sciex 4800 Plus MALDI time of flight/time of flight (TOF)/TOF Analyzer and confirmed with a Shimadzu MALDI-TOF 7090 mass spectrometer. An amount of 2 µL of the desalted sample (2.2 mg/mL) was mixed with 10 µL of the following matrix solution before spotting onto the insert: α-cyano-4-hydroxycinnamic acid (10 mg/mL) (CHCA, Fluka), acetonitrile (30%), and trifluoroacetic acid (0.1%).

Peptide mass fingerprint analysis. For peptide mass fingerprinting, the gp145 protein was analyzed with and without PNGase F prior to trypsin digestion [25]. The mMass program [28] and the Mascot tool were used to determine the peptide’s identity (https://www.matrixscience.com/, accessed on 19 September 2020).

Glycan analysis. PNGase F was used to release glycans in gel for glycan analysis [27,29]. The data presented were obtained using the 2′,4′,6′-trihydroxyacetophenone monohydrate (THAP) matrix, which could be used in either positive or negative mode and ionized the glycans better than super 2,5-dihydroxybenzoic acid (sDHB). Both matrices, THAP and sDHB, yielded similar results (data not shown). To increase the signal, the drops were spotted one after the other. The glycans were examined in both positive and negative reflector modes for neutral glycans and acidic glycans. An in-house Python program was developed to calculate the relative abundance of the glycans in the following manner in order to select the peaks of the mass spectra. First, the monoisotopic peaks were identified for the most prominent glycan peaks. Then, the other isotopic peaks were identified for each glycan (±1 Da, ±2 Da, etc.). Last, the ions for all peaks for one glycan were added. Finally, all the totals for all the glycans of interest were added to obtain the total ions count. The relative abundance was determined by dividing the total ions for a given glycan by the total ions value. The column graph and raw data images were prepared using GraphPad Prism (Version 6.07) and the program GlycoWorkbench. The values shown are the average of three readings and the error bar represents the standard error of the mean.

### 2.7. Mass Analysis

The 2100 Bioanalyzer (Agilent Technologies) microchip capillary gel electrophoresis system was used because it is a quick method for precisely determining the mass of biomolecules. As recommended by the manufacturer, 2 µg of gp145 protein was labeled in a 5 µL reaction volume and 7 ng of gp145 was loaded into one well. Agilent’s high-sensitivity protein 250 kit was employed. Data were analyzed using the 2100 Expert Software v B.02.08.SI648(SR2) and 1X DPBS were used as a diluent and to fill empty microchip wells when necessary.

### 2.8. Glycoform Profiling by Imaged Capillary Isoelectric Focusing (cIEF)

An amount of 70 μg of gp145 protein was mixed in 100 µL of cIEF solution urea (3 M), methylcellulose (0.35%), Servalyte 2–11 (1.8%), Servalyte 3–7 (1.8%), and phosphoric acid (8 mM). The focusing was done in two steps—the first step was run for 1 min at 1500 V and the second step was run for 7 min at 3000 V. Each sample was run in triplicate. The iCE3 system (ProteinSimple) with the PrinCE autosampler was used at 10 °C. The electropherogram was calibrated using the iCE3 21 CFR Part 11 software V4.1 (ProteinSimple) and plotted using GraphPad Prism (Version 6.07). For group analysis, the pI ranges were defined as follows: Group I (3.68–5.45), Group II (5.45–6.25), and Group III (6.25–7.50). Values for the group analysis are the average of three runs and the error reported is the standard deviation.

### 2.9. Size Distribution Analysis by Dynamic Light Scattering (DLS)

The DLS analysis was carried out using Sarstedt cuvettes (reference number: 67.754) and the Zetasizer Nano Zs (model number ZEN3600, Malvern Panalytical). Samples were diluted to 0.15 mg/mL in 1X DPBS. After being spun, samples were loaded into the cuvette. ICN PBS tablet with a dispersant refractive index of 1.330, temperature of 25 °C, and viscosity of 0.8882 was selected as the dispersant. For data analysis, the accompanying software was employed. The values reported are the average of three measurements and the error reported is the standard deviation.

### 2.10. Secondary Structure Analysis by Circular Dichroism

Data were collected using a 1-mm Jasco cuvette and a J-1500 circular dichroism (CD) spectrometer (Jasco). To acquire data, the following parameters were used: CD scale = 200 mdeg/1.0 dOD; D.I.T = 4 s; bandwidth = 1.0 nm; pitch = 0.1 nm; and scanning speed = 50 nm/min. In 0.1X DPBS, the gp145 glycoprotein was used at a concentration of 100 µg/mL. For data analysis, the accompanying software was used.

### 2.11. Synthesis of Supramolecular Hacky Sacks (SHS)

All commercially available reagents were used as synthesized and provided by the manufacturers. The synthesis procedure for ImAG is published elsewhere [30,31]. Briefly, samples, termed as Supramolecular G-quadruplexes (SGQ), were prepared by dissolving ImAG (0.650 mL, 5 mM) in potassium thiocyanate (KSCN) at 800 mM. The solid components were dissolved using phosphate buffer saline (PBS) (pH 7.4) in a 7-mL glass vial. The SGQ samples were sonicated for ~1 min and left equilibrating in the refrigerator overnight.

### 2.12. Fluorescent Labeling of gp145

The gp145 labeling and bioconjugation reaction was carried out in a 4 °C cold room according to the manufacturer’s protocol (CAS #: 3326-32-7). For the reaction, the starting concentration of gp145 was 0.6 mg/mL. The following was the procedure: In a 4-mL borosilicate vial, 1 mL sodium carbonate buffer (0.1 M, pH 9) was mixed with 1 mL gp145. Fluorescein isothiocyanate (FITC) was used as a labeling agent in anhydrous dimethyl sulfoxide (DMSO) at a concentration of 1 mg/mL. Next, 50 µL of FITC reagent was slowly added to each mL of protein sample using a mechanical micropipette. The reaction was stirred in the dark for 8 h. Subsequently, the protein was purified by removing unconjugated FITC using size-exclusion chromatography (PD10 column) using 1X PBS (pH 7.4) as the eluent. One-milliliter fractions of FITC labeled gp145 were collected in borosilicate vials. Afterward, the FITC-labeled protein fractions were pooled and concentrated by centrifugation using 10 mL AMICON filter tubes (30 kDa MWCO) spinning at 3900 rpm for 30 min. The final concentration of FITC-labeled gp145 (gp145-FITC) was measured using UV-VIS spectrophotometry.

### 2.13. Preparation of the gp145-FITC-SHS Complex and VRC01-Alexa Fluor 647 Complex

A stock solution containing ImAG (5 mM) and KSCN (800 mM) in 1X PBS was made fresh and left at −20 °C for 1 day. A second solution, termed SHS solution, containing ImAG (0.3 mM) and KSCN (52 mM) was prepared by diluting 100 µL of the stock solution in 1.650 mL of PBS at 40 °C. Then, 50 µL of the SHS solution was added to a 2 mL borosilicate vial and mixed with 50 µL of gp145 (0.282 mg/mL) in triplicates. These samples were then left to incubate spinning (100 rpm) at 40 °C for 1 h, then examined by flow cytometry and confocal microscopy. The fluorescent gp120, HIV-1 env gp120, strain IIIB, recombinant (FITC) was from MyBioSource (cat. # MBS635758).

### 2.14. Flow Cytometry

Flow cytometry measurements were made using a BD Acurri^TM^ C6 Flow Cytometer (BD Sciences) with standard laser configuration (488 nm and 640 nm) and detectors (FL1 525/25, FL2 585/15, FL3 670 LP, and FL4 675/25). Samples were analyzed quickly after the encapsulation protocol was finished in the incubation vial.

### 2.15. Confocal Imaging

Fluorescent images were captured at the Neuroimaging & Electrophysiology Facility in the Molecular Science Center (https://www.nief-upr.com/, accessed on 27 January 2023). The microscope utilized is an inverted Nikon Eclipse Ti with a 100X oil immersion Plan Apo λ objective. Stimulation and acquisition were performed by a Nikon A1 Laser Scanning Confocal unit using (excitation/emission) 488 nm/525 nm for FITC, 640 nm/700 nm for Alexa-647, and a differential interference contrast (DIC) configuration for transmitted light. The pinhole aperture was open to 0.8 airy and intensity profile analysis was performed on NIS Elements Software V5.3.

## 3. Results

### 3.1. Optimizing the Expression of gp145

The CHO-K1 cell line expressing secreted gp145 CO6980v0c22 was transferred to our facilities. Before its arrival, this cell line was cultured in 5 L bioreactors containing PowerCHO-2 chemically defined media. The culture supernatants gave a titer of 40 mg/L as measured by ELISA, according to the MTA. In the current work, we first examined eight different media bases with various supplementations (Appendix A) and confirmed that PowerCHO-2 was the best media base to grow our CHO-K1 cell line. We observed that transferring production from shake flasks to a bioreactor resulted in a substantial decrease in the gp145 production titer from 91 mg/L in shake flasks down to 47 mg/L in the bioreactor (Appendix A). To overcome this, we analyzed media pH at the end of our SF runs, and it was pH 6.8. To confirm pH 6.8 as optimal for cell growth and gp145 production, we performed small-scale bioreactor runs at different pH set points. Results confirmed that, in bioreactors, a pH of 6.8 allows gp145 titers (86 mg/L) similar to those obtained with SF runs (91 mg/L) (Appendix A). Since pH 6.8 resulted in the highest gp145 titer, we used this pH in subsequent runs. This change in pH set point allowed the cultures to remain productive for a longer time (Figure 1a,b), exhibiting viable cell density (VCD) values that remained above 80% during the first 8 days and then VCD went down to 45% on day 11 when the harvest was made (Figure 1a). These high VCDs observed during the first 8 days were accompanied by a high number of viable cells: 98.5 (day 1)–84.5% (day 8) (Figure 1b), which eventually decreased without losing their productivity. This extension of cell culture time at pH 6.8 correlated with a prolonged decrease in glucose concentration in the bioreactor (Figure 1c), together with a reduction of lactate accumulation after day 5 of culture (Figure 1d). These data suggest that our CHO-K1 cells, at pH 6.8, are able to co-metabolize glucose and lactate during the growth phase at day 5 when lactate accumulation was reversed (Figure 1d), and the VCD still had not reached its maximum cell density at day 7 (Figure 1a). Monitoring of metabolites showed that from the third day, the glucose levels decreased constantly until almost completely consumed by day 7 in a process independent of the pH of the cell culture (Appendix A). This glucose consumption coincides with the decrease in the concentration of GlutaMAX (Appendix A) as well as with the accumulation of ammonia in it (Appendix A). From the fifth day, the accumulation of ammonia reached a plateau without significant changes until the eleventh day of culture monitoring (Appendix A).

The production of gp145 was scaled up to a 50 L bioreactor (40 L working volume) batch culture (Figure 2a). Cell growth peaked at day 8 with a viable cell density of 8.3 × 10^6^ cells/mL and a viability of 98%. Although the cells began to decline in viability from day 9 onward, the growth was allowed to proceed since titers continued to increase without a substantial increase in the amount of CHO host cell proteins (HCPs) (Figure 2b). During the 11 days of cell culture in this 50 L bioreactor, a time-dependent accumulation of ammonia was detected (Figure 2c). Said accumulation did not reach the plateau observed in the 0.250 L bioreactor, instead, it exhibited a sustained increase until harvest (day 11). For GlutaMAX, similar to what was observed in the 0.250 L bioreactor on day 5, the levels reached practically undetectable levels, demonstrating the consumption profile of the cells (Figure 2c). Additionally, on day 5, lactate levels began to decline while lactate dehydrogenase (LDH) levels consistently spiked through the harvest day. Glucose levels were depleted after 9 days (Figure 2d). In theory, increasing the harvest time from 7 days to 11 days carries the risk of excessive cell death, potentially resulting in the release of large amounts of HCPs into the supernatant. Our antibody-based biosensor assay showed that while the gp145 titers were increasing linearly until day 11, the abundance of HCPs in the supernatants remained relatively constant, further supporting the strategy of extending the lifetime of the culture (Figure 2b). The clarified harvest sample gave a yield of 101 mg/L.

We tested supplementation of PowerCHO-2 with ornithine, citrulline, and chemically defined lipids without an effect on titers (data not shown). To further optimize the production of gp145, we explored the effect of a continuous feed of nutrients as well as GlutaMAX supplementation. In Figure 2c, we identify that both glutamine and GlutaMAX were limiting as the levels of both metabolites reached concentrations below the detection limit after 5 days in culture while ammonia accumulation increased rapidly from day 2 and LDH production maintained an increasing rate from day 6. Glucose consumption remained constant until the glucose levels were below the detection limit. As shown in Appendix A, we explored the effect of adding GlutaMAX in the glucose feed strategy to supplement the culture under three conditions. The objective of the first condition was to feed on particular days (day 3 and day 6) to replenish glucose to 6 g/L and GlutaMAX to 4 mM. The second condition consisted of adding glucose and GlutaMAX in combination to maintain glucose levels between 2–3 g/L, with feedings on days 5, 7, and 11. The objective of the latter was to maintain those levels, not to feed on particular days. The third condition was to supplement with glucose only up to 6 g/L. From the results obtained, it is evident that the second treated condition, which consists of maintaining glucose levels between 2–3 g/L, produced the best titers (Appendix A).

### 3.2. Analysis of Whole gp145 by SDS PAGE and MALDI-TOF Mass Spectrometry

The yield of our purified gp145 was 1 mg from 1 L of supernatant. To confirm that the gp145 produced was similar in terms of molecular weight to the reference standard (RM), we performed an SDS-PAGE analysis. The purified gp145 ran as expected, with a band corresponding to a 145 kDa protein (Figure 3a). A complementary approach for the determination of the molar mass of gp145 was MALDI-TOF mass spectrometry of the intact full-length protein (Figure 3b). The results showed a molar mass of 128,600 for the (M + H)^+^ ion with an additional signal of 64,600 corresponding to the (M + 2H)^2+^ ion, similar to other published data [32]. According to this mass spectrometry result, the contribution of the glycan shield to the molar mass of gp145 is 50,000 Da or roughly 40% of the total mass. A third method employed for the mass analysis of gp145 was the microchip capillary gel electrophoresis (MCGE) system, which provides higher sensitivity than traditional gel electrophoresis. MCGE analysis showed that gp145 migrates as a 310 ± 5 kDa protein, which is outside of the linear range for the method, which has an upper limit of 250 kDa (Figure 3c). The substantial decrease in the electrophoretic mobility observed by MCGE has been documented for other extensively glycosylated proteins [33]. Although the exact causes for this electrophoretic delay are not known, it is thought to originate from a lower overall protein charge, since the extensive glycan shell on the protein limits its covering with detergent molecules [33]. Nonetheless, this method can still be utilized to determine the purity of the samples (Appendix A).

These methods clearly show that our purified gp145, reported here, has similar attributes to the reference standard. Furthermore, both materials were confirmed to be the same amino acid sequence by peptide mass fingerprinting (PMF), as shown in Appendix A.

### 3.3. Glycan Analysis

To show that the parameters adopted to increase gp145 titers did not affect the overall *N*-glycosylation profile of the vaccine candidate, we performed glycan analysis by MALDI-TOF mass spectrometry. First, gp145 was treated with PNGase F to determine how the removal of glycans affects the peptide mass fingerprinting. Removing the *N*-glycans with PNGase F resulted in a 2-fold increase in peptide coverage (from 18% to 38%), mostly due to an increased abundance of unmodified peptides in the analysis (Appendix A). Then, the gp145 PNGase F-released glycans were analyzed by MALDI-TOF. The neutral glycan profile was similar to that published by Doores et al. [34] (Figure 4 and Appendix A). Likewise, the acidic glycan profile coincides with glycans previously identified by other groups [35] for HIV Env (Appendix A). The glycosidic linkage structure of the most prominent peak, mass 2076.6, was confirmed by collision-induced dissociation (data not shown).

### 3.4. Biochemical Analysis of gp145 by Capillary Isoelectric Focusing (cIEF) and RP-HPLC

Although the predicted isoelectric point (pI) for gp145 polypeptide alone is above 7.0, the acidic glycans shift the gp145 pI and impart substantial charge heterogeneity. We analyzed the distribution of charged isoforms by capillary isoelectric focusing (Figure 5a). The pI distribution spans a range from 4.3 to 7.2, which is consistent with other pI determinations reported for gp120 [36]. Treatment of gp145 with neuraminidase A resulted in a substantially increased pI, indicating that the native pI distribution is highly influenced by the presence of sialic acids [27]. The pI distribution for the gp145 material made in our facility was very similar to that of the reference standard, indicating similar levels of sialylation (Appendix A). The RP-HPLC hydrophobic footprint analysis shows that the gp145 made in our UPR-CBC facility, is nearly identical to the RM (Figure 5b). The retention times were identical for our purified gp145 and the reference standard, namely, 5.326 ± 0.002 min. This retention time of less than 10 min was critical in developing a method that could be implemented into a quality assay. Furthermore, the tail factors were 1.008 ± 0.005 for our in-house produced gp145 and 1.130 ± 0.008 for the RM, suggesting a homogenous product and a peak shape consistent with the requirements for a quality control setting. The samples had a percent recovery of 106% and 107% for the RM and gp145, respectively; the standard curve used is shown in Appendix A.

### 3.5. Oligomeric State of gp145

An important consideration in making new versions of Env longer than gp120 is the preservation of the native trimeric quaternary structure [4,9]. Here, we report two different assays by which the oligomeric state of gp145 can be monitored. Dynamic light scattering (DLS) analysis shows that our gp145 protein is moderately (0.1–0.4) [37] polydisperse because it has a polydispersity index (PDI) of 0.22 (Appendix A, Appendix A), suggesting that it could be a mixture of different populations of gp145. This finding is consistent with the results obtained from native gel electrophoresis (Figure 6), where four different populations of gp145 were observed, namely, higher order aggregates, trimer, dimer, and monomer. The observed distribution of trimers, dimers and monomers was exactly the same as that reported previously by Wieczorek et al., since the electrophoresis was performed using the same MW standards as in the cited reference [11]. In that work, the assignment of bands was consistent with the negative stain EM images, revealing the true oligomeric states represented on the gel [11]. Interestingly, our gp145 seems to have less aggregate and more trimers than the reference standard. Size exclusion chromatography showed similar results (Appendix A).

### 3.6. Native-Like Properties of gp145

To assess the native-like properties of gp145, namely the secondary structure, we analyzed the protein by circular dichroism (CD) [38]. A broad absorption band with a minimum at 210 nm and an additional signal at 220 nm reveals the presence of a well-structured protein that is spectroscopically similar to the RM (Figure 7a). Further confirmation of the native-like functionality of gp145 was evidenced by the binding affinity to the broadly neutralizing antibody 4E10, which was measured using bio-layer interferometry (BLI) (Figure 7b). The control for the experiment is presented in Appendix A. The K_D_ values were 46 nM and 65 nM for the reference standard and for our purified gp145, respectively (Appendix A).

### 3.7. Encapsulation of gp145 for Delivery

In addition to the patent need for new immunogens, there is an increased drive to develop novel delivery platforms such as osmotic pumps and nanoparticles to ensure a sustained release of immunogenic proteins. Toward this end, we tested self-assembling supramolecular hacky sacks (SHS) microparticles for their ability to encapsulate the gp145. Previously, it was demonstrated that our SHS microparticles increased the frequency of IFN-γ-producing cells in a murine model independently immunized with pGag and pA27L plasmids [39]. The delivery of DNA plasmids coding “wild-type” antigens using these SHS microparticles efficiently increased antigen-specific humoral and cell-mediated immunity in mice. To validate that these microparticles are capable of successfully encapsulating highly glycosylated proteins we first proceeded to encapsulate FITC labeled gp120 (gp120-FITC). Results obtained by flow cytometry show that the microparticles encapsulated gp120-FITC, a response that can be seen as a significant shift (96.4%) to the right in the cytometric histograms (Figure 8a). The cytometric analysis focused on detecting FITC showed a significant increase (**** *p* < 0.0001) in the microparticles that incorporated gp120-FITC, compared to those not incubated with the protein (Figure 8b). Similar responses were obtained for gp145-FITC. Thus, our SHS microparticles served to favorably encapsulate the highly glycosylated and heavy trimeric gp145. Said response can be appreciated by the notable displacement (68.3%) to the right in the histogram of the cytometric analysis (Figure 8c). The encapsulation of gp145-FITC was statistically significant (**** *p* < 0.0001) (Figure 8d). Altogether, the SHS microparticles prove to be a reliable vehicle for encapsulating and delivering our highly glycosylated antigen, gp145.

To understand the details of gp145-FITC encapsulation, SHS microparticles were examined by confocal microscopy. Figure 9a shows 3D fluorescent and differential interference contrast (DIC) images of microparticles containing gp145-FITC (left) and microparticles alone (right) with a size of ~2000 nm (~2 µm). A 2D image shown was used to determine the distribution of the gp145-FITC envelope across the microparticles (Figure 9b). The intensity of the gp145-FITC was homogeneously dispersed within microparticles (Figure 9c). Figure 9d shows two distinct microparticles of ~1600 nm, one loaded with gp145-FITC (left) and a second one that was first loaded with gp145-FITC followed by incubation with VRC01 broadly neutralizing the antibody [40] and subsequently incubated with a secondary antibody (Alexa Fluor 647) recognizing the VRC01 (acting as primary antibody). The broadly neutralizing antibody VRC01 binds strongly to monomeric gp120, which is known to cover 98% of the target site, comprising 1089 Å^2^ on the gp120 outer domain—about 50% larger than the 730 Å^2^ surface utilized by CD4 [41]. The 2D image shown in Figure 9d was generated to determine the distribution of intensities of the gp145-FITC and Alexa Fluor 647-VRC01-gp145-FITC complex across the microparticle shown in Figure 9e. The analysis of these images reveals that Alexa Fluor-647-VRC01-FITC-gp145 was distributed across the microparticles; however, unexpectedly, the highest intensities were located at the surface of these microparticles (the VRC01 binding towards the microparticle is negligible, Appendix A).

## 4. Discussion

Manufacturing is a major barrier in the development of protein-based HIV vaccines. HIV glycoproteins present unique manufacturing challenges due to significantly lower production titers than other biological products [14]. The high level of glycosylation of HIV Env, combined with its wide charge heterogeneity, makes defining critical quality attributes difficult. The successful transfer and optimization of an upstream process for the production of trimeric gp145 is reported here. We extended the life of the culture and increased protein production from 40 mg/L to 70 mg/L in shake flasks by systematically adjusting the media composition, pH, and feeding schedule. Further up-scaling of the upstream protocol into bioreactors resulted in an immediate decrease in the production titers down to 40 mg/L when the pH was maintained at 7.2. However, changing the pH set point to 6.8 (the measured pH in shake flask cultures) prolonged cultivation time and resulted in higher titers. Others have reported higher production titers for Epo-Fc [42] and a monoclonal antibody [43] by maintaining the pH at 6.8. The authors discovered that keeping the pH at 6.8 resulted in a delayed metabolism marked by a 2-fold decrease in glucose consumption and a decrease in glutamine consumption [42]. Maintaining a pH of 6.8 has been shown to increase titers without affecting cell-specific productivity in the production of monoclonal antibodies [43]. We show that keeping the pH at 6.8 can extend the cell culture’s lifetime to 11 days, resulting in higher titers without significantly increasing the number of host-cell proteins, even with decreasing viable cell densities.

Following the RV144 vaccine trial in Thailand, which used gp120 as the boost immunogen, numerous updated versions of the Env glycoprotein were created and tested in animals and humans [1]. The first cGMP method for producing native-like trimers of BG505 SOSIP gp140 in a 200 L bioreactor reported a final yield of 17 mg per liter of culture [23]. Given that this construct requires the formation of an intramolecular disulfide bond as well as enzymatic cleavage by co-expressed human furin protease, the reported yield for the BG505 SOSIP is high. The authors report that they harvested the supernatants on day 15 and obtained production titers ranging from 85 to 165 mg/L, which were then purified using antibody-based affinity purification. Another protocol for producing uncleaved gp145 in shake flasks was recently reported, with production titers of 47 mg/L after several optimization steps. The authors discovered that extending the culture time beyond the first 7 days, combined with a temperature change to 32 °C, resulted in higher titers despite decreasing viable cell density [22]. In the current work, we combine several of these optimal parameters to achieve even higher titers of uncleaved gp145 in a 1 L bioreactor of 160 mg/L. Remarkably, we obtained higher titers for gp145 using our method than had previously been reported [22].

The inherent heterogeneity of biological products poses a challenge to traditional notions of product quality. In the case of HIV Env, the extensive glycosylation [44,45] with variable glycan occupancy and variable sialylation results in a wide charge isoform heterogeneity. Several methods in our analytical toolbox have been specifically developed to address heterogeneity as a potential indicator of quality. The glycosylation of Env was investigated in the current study using MALDI-TOF analysis of PNGase F-released glycans, which revealed that the gp145 produced in our facilities is identical to the glycan distribution of the reference standard. By imaging capillary isoelectric focusing, we further demonstrated that the charge distribution of our gp145 was comparable to that of the reference standard. Despite having a basic amino acid sequence, gp145’s experimentally determined isoelectric point distribution is acidic because sialic acid has been heavily incorporated into the glycan shield.

A quality-by-design paradigm has been steadily developing in the design of pharmaceutical and vaccine manufacturing processes, necessitating close attention to key quality attributes throughout the process [46,47]. In order to achieve this, biotech companies advise using multi-attribute methods (MAMs), which use a single platform—typically high-resolution mass spectrometry—to interrogate multiple parameters from a single sample simultaneously or in parallel. The LC/MS/MS platform is the technique of choice for MAM development because it allows for simultaneous monitoring of protein integrity and post-translational modification. Here, we present the use of MALDI-TOF MS as an alternative multi-attribute platform. We get the peptide mass fingerprint of gp145, an analysis of released glycans in positive and negative reflector modes with orthogonal signals for high-mannose and sialic acid-containing glycans, and an accurate molar mass for the entire protein from this instrument. The MALDI-TOF platform, as a multi-attribute platform, could be used in place of SDS-PAGE gel and capillary electrophoresis for protein size and purity, with additional information on protein degradation and glycosylation.

The recent use of lipid nanoparticles for immunization has opened up a new avenue for testing other non-lipid particles as a strategy for delivering immunogens. To that end, we encapsulated gp145 in this study using proprietary guanosine microparticles [31]. The size of the microparticles containing gp145-FITC is between 1000 and 2000 nm. Importantly, particle size determines the size-dependent mechanism of lymph node trafficking [48]. Smaller nanoparticles (20–200 nm) travel more efficiently to draining lymph nodes in a rodent model than larger particles (500–2000 nm) [48]. Large microparticles, such as the gp145-SHS complex, will be dependent on their interaction with dendritic cells, according to the study. Nonetheless, by controlling the self-assembly process, the size of the SHS microparticles can be significantly reduced. We have prepared 400 nm SHS for use in future experiments. We are also determining the optimal spacing and density of gp145 trimers in these microparticles, as these are critical factors in B cell activation. As we detected the unexpected binding of VRC01 on the surface of the SHS microparticles, we speculate that they may serve not only for delivery but also for antigen presentation or as an adjuvant. The field of particle-based delivery systems agrees that effective systems must strike a balance between safeguarding an antigen, such as gp145, from degradation and efficiently presenting it on the surface towards antigen presenting cells (APCs) [49]. Our research group found that SHS particles can improve both arms of adaptive immunity in experimental plasmid-based vaccines in mice, compared to antigens without particles [39]. Although pDNA and protein-based vaccines have different mechanisms, both require accessible antigens to confer effective immunity. The UPR-CBC [25] will now conduct a preclinical trial to assess the gp145-SHS complex as an immunogen.

## 5. Conclusions

Here, we have described how UPR-CBC implemented an expression scheme in our facility as a part of a technology transfer agreement. Furthermore, we doubled the titers from the technology transfer in shake flasks. As a means to scale up the production, we started with a 0.250 L bioreactor and found that the previously validated shake flask conditions did not translate into higher yields in a bioreactor. In a study to optimize the pH of the growth, we found that pH 6.8 in 0.250 L bioreactors increased titers to 86 µg/mL. To scale-up the process further, we used different-sized bioreactors, starting at 1.5 L and moving up to 5 L and 10 L, and ultimately 40 L, to demonstrate how we can scale up the process with continued higher titers, namely 101 mg/L (Figure 10). This figure depicts the scaling process used in this project, wherein we successfully transferred our cell culture to larger volume vessels employing various feeding strategies. In summary, we demonstrate that UPR-CBC has established a robust analytical pipeline for the development, production, and optimization of gp145-SHS, paving the way for HIV vaccine candidates to be manufactured in accordance with cGMP for preclinical and clinical trials. More importantly, we present a new patented microparticle with encapsulating properties that enables glycoproteins to be accommodated within and expressed on its surface. Our findings are especially relevant in light of the recent clinical trial failures for Uhambo (HVTN 702) [50], Imbokodo (HVTN 705) [51], and, most recently, MOSAICO (HVTN 706) [52]. Despite being safe, all three candidates did not significantly reduce the risk of HIV infection. It is now more important than ever to establish novel platforms for the development of new vaccine candidates, together with innovative delivery systems, that are readily manufacturable. Because the need for an HIV vaccine remains, it is critical to test new candidates in new delivery platforms as soon as possible. Our new microparticle is part of these new HIV vaccine development efforts.

## Figures and Tables

**Figure 1 vaccines-11-00975-f001:**
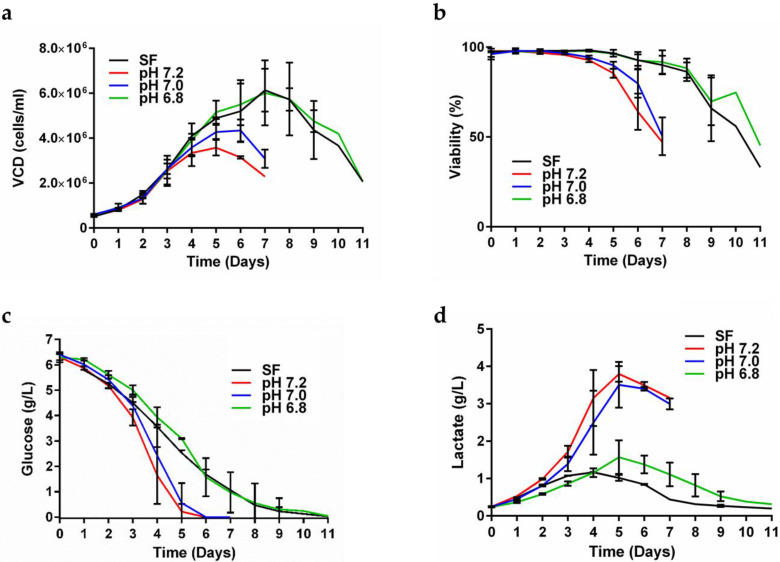
The pH effects on CHO-K1 cell growth. Cells were grown in a 0.250 L bioreactor at different pH set points. (**a**) Assessment of bioreactor viable cell density (VCD) of cells grown at different pHs. (**b**) Percent viability. (**c**) Glucose consumption profile. (**d**) Lactate production/consumption of cells grown at different pHs. The data presented are the average of two runs and the error bars represent the standard deviation. On days 10 and 11, a single bioreactor was left until day 11. The shake flask (SF) was run as a control (black). In all figures Red = pH 7.2; Blue = pH 7.0; Green = pH 6.8.

**Figure 2 vaccines-11-00975-f002:**
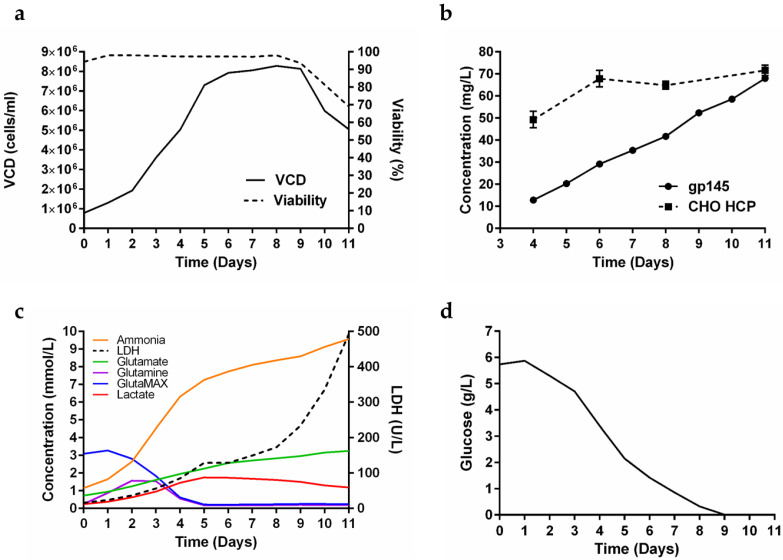
Production of gp145 in a 40 L bioreactor. (**a**) Assessment of VCD and percent viability over the 11 days from seeding to harvest for the growth. (**b**) Daily titer determination by RP-HPLC and CHO host cell proteins (HCPs) quantification. The titer values are the average of three readings. The HCP values are an average of 12 readings. In both graphs, the error shown is the standard deviation. (**c**) Profile of other metabolites for the growth (red = lactate; orange = ammonia, (NH_3_); green = glutamate; blue = GlutaMAX; purple = glutamine; dashed line = lactate dehydrogenase, LDH). (**d**) Glucose profile of the growth. The lower detection limits of the instrument are: glucose (0.02 g/L), glutamate (0.1 mmol/L), glutamine (0.01 mmol/L), GlutaMAX (0.01 mmol/L), NH_3_ (0.278 mmol/L), lactate (0.0444 mmol/L), and LDH (20 U/L).

**Figure 3 vaccines-11-00975-f003:**
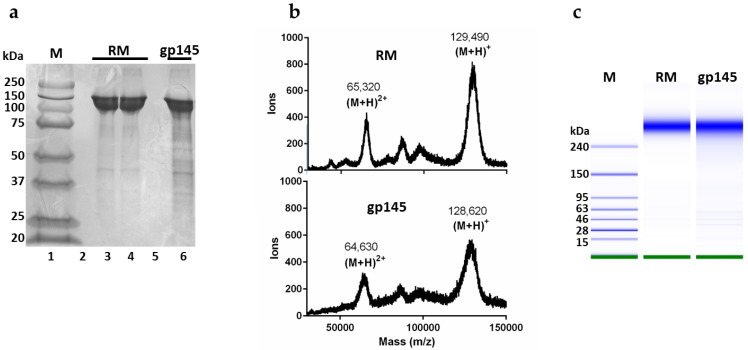
Molecular mass determination of HIV gp145 by SDS-PAGE, MALDI-TOF, and MCGE. The gp145 from the 50 L bioreactor was analyzed. (**a**) For SDS-PAGE analysis, 10 µg of gp145 produced in CHO cells was applied. (**b**) The MALDI-TOF whole protein analysis was performed using the CHCA matrix in linear positive mode. (**c**) The MCGE analysis of our gp145 (7 ng) was carried out with the High Sensitivity Protein 250 kit, to produce a gel-like image representative of the run. In panel C, the M column represents protein markers. RM = reference standard.

**Figure 4 vaccines-11-00975-f004:**
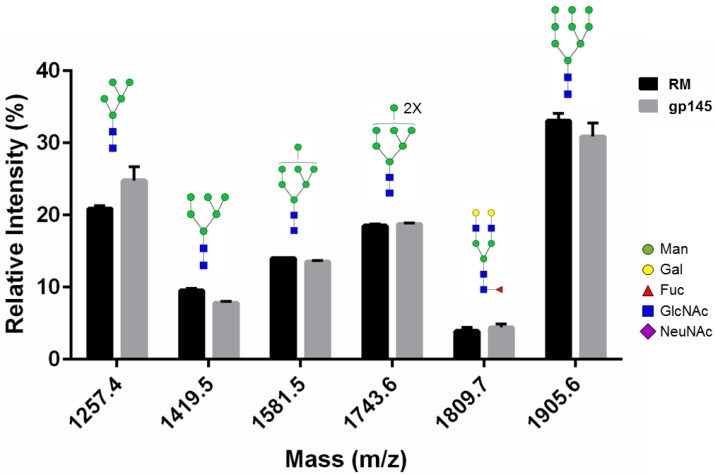
Glycan analysis of the *N*-linked glycans released from HIV gp145 protein. This glycan profile shows the relative abundance of the most abundant neutral glycans identified by positive reflector MALDI-TOF analysis. The recombinant gp145 (10 µg) was resolved through SDS-PAGE gel and digested with PNGase F. Recovered *N*-glycans were co-crystalized with THAP matrix. The predicted glycan structures are shown, from left to right, oligomannose-5, oligomannose-6, oligomannose-7, oligomannose-8, G2F, and oligomannose-9. Error bars represent SEM. RM = reference standard.

**Figure 5 vaccines-11-00975-f005:**
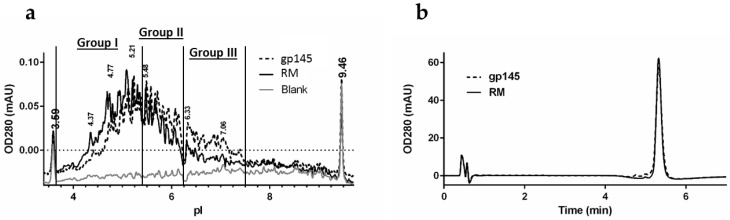
Biochemical characterization of gp145. (**a**) Representative glycoform pI profile by cIEF of HIV gp145 protein. Buffer was used as a blank control. The pI markers 3.59 and 9.46 were used to calibrate the electropherograms. (**b**) Representative trace of the RP-HPLC hydrophobic footprint analysis of gp145 Env. RM = reference standard.

**Figure 6 vaccines-11-00975-f006:**
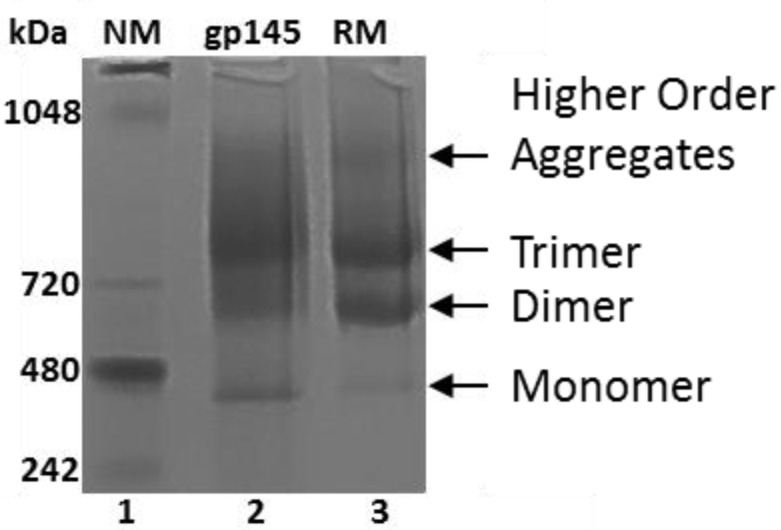
Oligomerization analysis of gp145. The gp145, 10 µg, (lanes 2 and 3) analysis by native PAGE. The NativeMark (NM) protein standard was used in lane 1 (Invitrogen Novex). RM = reference standard.

**Figure 7 vaccines-11-00975-f007:**
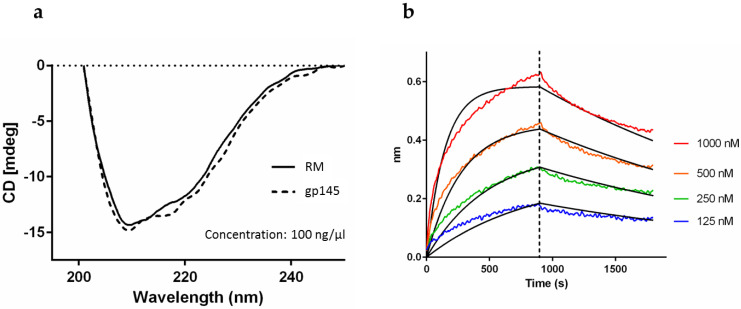
Integrity analysis of gp145. (**a**) CD analysis of gp145. The protein was diluted in 0.1X DPBS. RM = reference standard. (**b**) Bio-layer interferometry analysis of gp145. Our gp145 binds the broadly neutralizing antibody, 4E10, with an affinity of 65 nM.

**Figure 8 vaccines-11-00975-f008:**
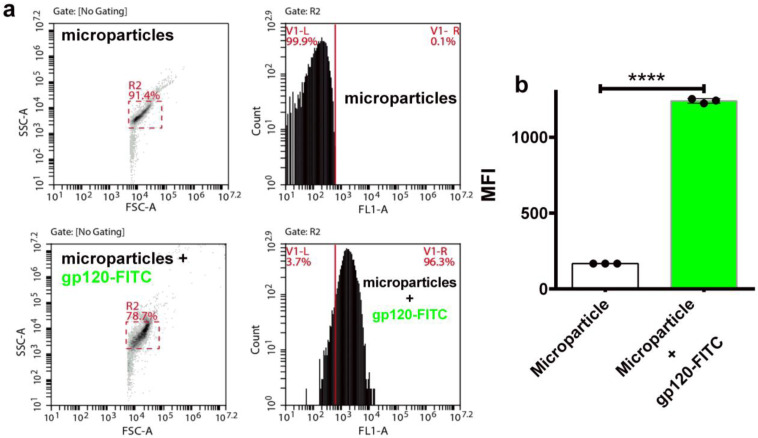
Encapsulation of gp145 in SHS microparticles. (**a**) Microparticles were evaluated by flow cytometry. The gp120-FITC glycoprotein was incubated and encapsulated in the microparticles (lower panels). (**b**) The observed encapsulation was statistically significant. (**c**) Microparticles were evaluated by flow cytometry before being used to encapsulate the antigen (top panels). In lower panels, microparticles containing gp145-FITC show a right shift toward higher values of fluorescence. (**d**) Encapsulation of gp145-FITC was effectively achieved as determined statistically. **** *p* < 0.0001, unpaired *t* test.

**Figure 9 vaccines-11-00975-f009:**
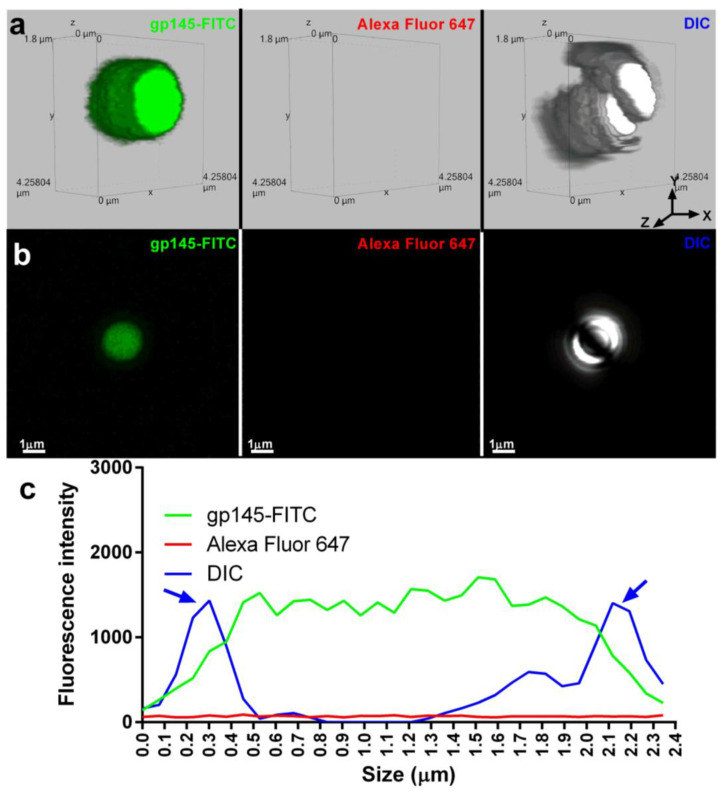
SHS microparticles encapsulate and present gp145 on their surface. (**a**) A representative gp145-FITC-containing microparticle is shown on the left. The lack of Alexa Fluor 647 signal in the middle panel microparticle indicates that there is no autofluorescence at the analyzed wavelength. A representative DIC image of microparticles is shown on the right. (**b**) The fluorescence emitted by the encapsulated gp145-FITC is depicted in 2D in the extreme left panel, the absence of fluorescence at the analyzed wavelength in the central panel, and the 2D DIC image in the extreme right panel. (**c**) A graph depicting the fluorescence emitted by gp145-FITC encapsulated (green line). In DIC, the microparticle is represented by a blue line with blue arrows indicating its borders. (**d**) A 3D representation of a microparticle encapsulating gp145-FITC (left panel). The binding of a secondary antibody conjugated to Alexa Fluor 647 that recognizes VRC01 and strongly binds to gp145-FITC is shown in the center panel. A representative 3D DIC image is shown (right panel). (**e**) A representative 2D image of encapsulated gp145-FITC is shown (far left panel). In the middle panel, VRC01 recognizes gp145-FITC fluorescence, while DIC capture is shown in the far-right panel. (**f**) Graph of encapsulated gp145-FITC (green line), gp145-FITC on the microparticle surface (red line), and the DIC corresponding to the microparticle (blue line).

**Figure 10 vaccines-11-00975-f010:**
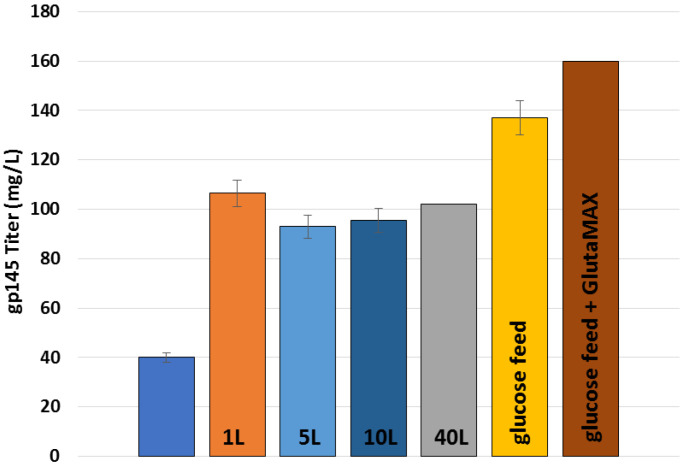
Summary of scale-up capabilities and increased titers of gp145. Technology transfer 5 L bioreactor reported titers, blue; 1 L bioreactor titers, orange (*n* = 7); 5 L bioreactor titers, light blue (*n* = 2); 10 L bioreactor titer, dark blue (*n* = 5); 40 L bioreactor titer, grey (*n* = 1); 1 L glucose feed, yellow (*n* = 5); and 1 L glucose feed plus GlutaMAX, brown (*n* = 1).

## Data Availability

The data supporting the conclusions of this study are reported in the article and the Appendix A. Data are available from the corresponding author upon reasonable request.

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
