# Peer review of "Optimizing the Production of gp145, an HIV-1 Envelope Glycoprotein Vaccine Candidate and Its Encapsulation in Guanosine Microparticles"

_vaccines, 2023, doi:10.3390/vaccines11050975_

Round 1

Reviewer 1 Report

Akamine et al. optimized the production of gp145, a recombinant HIV-1 Env glycoprotein expressed in CHO-K1 cells, and scaled up its production. Using a series of sophisticated methods they showed that the N-glycosylation profile of their vaccine candidate was similar to that of a reference standard, and characterized its oligomeric state and secondary structure, too. Using proprietary guanosine microparticles, the Authors demonstrated successful encapsidation of FITC-labeled gp145 into such microparticles which may facilitate the delivery of distinct vaccine candidates. This is an interesting study and I recommend the publication of the manuscript after a revision.

Minor points:

line 110, …4.6x105…  - please change to: …4.6x105

line 780, reference 31. – perhaps the Authors may wish to indicate that this reference corresponds to a patent, according to:   https://patents.justia.com/patent/10106572

line 784, reference 33., Engel N. et al. - - perhaps the Authors may wish to indicate that this reference is actually an Application Note; as a matter of fact Engel et al. wrote about the same topic in a free PMC article, too:

Engel NY, Weiss VU, Wenz C, Glück S, Rüfer A, Kratzmeier M, Marchetti-Deschmann M, Allmaier G. Microchip capillary gel electrophoresis combined with lectin affinity enrichment employing magnetic beads for glycoprotein analysis. Anal Bioanal Chem. 2017 Nov;409(28):6625-6634. doi: 10.1007/s00216-017-0615-0. Epub 2017 Sep 20. PMID: 28932887; PMCID: PMC5670189.

 A major point:

in lines 412-414 the Authors refer to Figure 3c and write about the mass analysis of gp145 using MCGE. They claim that ”The substantial increase in the migration of this glycoprotein could be due to non-specific interactions between the surface glycans of gp145 and the gel matrix [33].” – Please rephrase that sentence, because there was no ”increase” in the migration of gp145 (on the contrary, a slower migration is apparent, I think), and in reference 33 Engel et al do not attribute altered migration of glycoproteins to non-specific interactions between the glycoproteins they tested and the gel matrix.

I recommend the publication of the manuscript after a revision.

Author Response

Response to Reviewer #1: 

Minor points: 

1) line 110: We have changed "4.6x 105" to "4.6 x 105". 

2) line 780: It has been more explicitly stated that this reference corresponds to a patent, as suggested. 

3) line 784: For reference 33, we have changed the Engel reference to the following: 

Engel N, Weiss VU, Wenz C, Rüfer A, Kratzmeier M, Glück S, Marchetti-Deschmann M, Allmaier G. Challenges of glycoprotein analysis by microchip capillary gel electrophoresis. Electrophoresis. 2015 Aug;36(15):1754-8. doi: 10.1002/elps.201400510. 

We agree with the reviewer that the previous reference was perhaps not the most appropriate for addressing the issue of delayed migration of glycoproteins in all electrophoretic methods, including MCGE. We believe this new reference more directly addresses the issue of the impact of glycosylation on electrophoretic mobility. 

Major points: 

4) The reviewer wrote the following: 

“in lines 412-414 the Authors refer to Figure 3c and write about the mass analysis of gp145 using MCGE. They claim that ”The substantial increase in the migration of this glycoprotein could be due to non-specific interactions between the surface glycans of gp145 and the gel matrix [33].” – Please rephrase that sentence, because there was no ”increase” in the migration of gp145 (on the contrary, a slower migration is apparent, I think), and in reference 33 Engel et al do not attribute altered migration of glycoproteins to non-specific interactions between the glycoproteins they tested and the gel matrix.” 

We agree with the reviewer that the extensive glycosylation of gp145 does not result in "increased migration". Thank you for pointing out this mistake. Instead, the extensive glycosylation of gp145 results in a delayed electrophoretic mobility, which in turn results in the overestimation of the molecular weight. In the manuscript we have substituted the sentence that reads: 

"The substantial increase in the migration of this glycoprotein could be due to non-specific interactions between the surface glycans of gp145 and the gel matrix" 

The updated text reads: 

"The substantial decrease in the electrophoretic mobility observed by MCGE has been documented for other extensively glycosylated proteins [33]. Although the exact causes for this electrophoretic delay are not known, it is thought to originate from a lower overall protein charge, since the extensive glycan shell on the protein limits its covering with detergent molecules [33]" 

This explanation is more consistent with prior observations on electrophoretic delay in glycoproteins made by Engel et al. We appreciate the reviewer pointing out this issue and helping to make it clearer in the manuscript. 

Reviewer 2 Report

The authors described a GMP pipeline to express, purify, and qualify an HIV envelope protein (Env) gp145 from Chinese Hamster Ovary cells. The yield and protein quality were impressive. Additionally, the authors presented a novel guanosine microparticle with gp145 encapsulated and displayed on its surface, which may be advantageous over soluble proteins as an immunogen. Overall, the data supported the conclusions.  It may be useful to describe exactly how the pH of the conditioned medium was adjusted and maintained.

1. What is the main question addressed by the research?

The authors described a GMP pipeline to express, purify, and qualify an HIV envelope protein (Env) gp145 from Chinese Hamster Ovary cells. Additionally, the authors presented a novel guanosine microparticle with gp145 encapsulated and displayed on its surface, which may be advantageous over soluble proteins as an immunogen.

2. Do you consider the topic original or relevant in the field? Does it address a specific gap in the field?

Yes, the topic is original and relevant in the HIV vaccine field.  GMP productions of soluble HIV envelope gp145 proteins have been challenging, and the manuscript addressed this gap of knowledge in the field.

3. What does it add to the subject area compared with other published material?

The GMP pipeline presented in the paper optimized the HIV gp145 protein expression. The yield and protein quality were impressive, much better than other published materials.

4. What specific improvements should the authors consider regarding the methodology? What further controls should be considered?

It would be useful to describe exactly how the preferred pH (6.8) for the conditioned medium was adjusted and maintained.

5. Are the conclusions consistent with the evidence and arguments presented and do they address the main question posed?

Overall, the data supported the conclusions.

6. Are the references appropriate?

The references are appropriate.

7. Please include any additional comments on the tables and figures.

The tables and figures are clear.

Author Response

Response to Reviewer #2: 

1) The reviewer wrote the following: 

“It would be useful to describe exactly how the preferred pH (6.8) for the conditioned medium was adjusted and maintained.” 

The main concern relates to the need for a clearer explanation of exactly how the pH was maintained at pH 6.8 in the optimized media. The pH of the media was adjusted inside the bioreactor vessel by sparging with CO2 until the pH reached 6.8. A sentence has been added to the Materials and Methods section (on page 3 under the Optimal pH Determination section) that reads:  

"The pH of the media was adjusted by sparging with CO2 at 0.6 SLPM prior to inoculation with 0.5 x 106 cells/mL"  

Furthermore, under the section Scale-up the following sentence was added:  

“Prior to inoculation, the pH of the media was adjusted by sparging with CO2 at 0.6 SLPM.” 

Reviewer 3 Report

In this study, Akamine et al. have endeavored to establish an optimal process for the large-scale production of HIV gp145 in CHO cells that is amenable to cGMP regulations for eventual preclinical and clinical vaccine trials. Following optimization of production titer in 50-L bioreactors, the authors have proceeded to extensively characterize the biophysical and biochemical attributes of their purified glycoprotein and, by doing so, demonstrate that it is virtually identical to a reference standard. Finally, using fluorescence and immunofluorescence approaches they show that the gp145 can be packaged into guanosine microparticles and further provide evidence for the elaboration of the glycoprotein on the surface of these particles. Overall, it is this reviewer’s impression that the study was very well conducted, the experiments presented have the proper technical controls, the manuscript is well written and the data presented support the authors’ conclusions that their optimized pipeline can produce decent titers of a properly glycosylated gp145 which can be packaged into their proprietary microparticles for delivery into target cells.

Major issues that this reviewer would like to see the authors address in their revision are as follows:

11) To provide clarification of the potential biological or immunological significance of the observed detection of gp145 on the surface of their encapsulated microparticles. Would this surface expression enhance the ‘delivery’ capability of these microparticles for gp145? With reference to the authors comments in Lines 551-552 and 640-641, would the surface expression of the glycoprotein be an important attribute to facilitating antigen presentation? If so, would these processes be less efficient with an internalized immunogen? Please clarify.

 2) The authors report their production titer (ie. that they are able to generate >100 mg/L of gp145) from the large-scale bioreactor. Can the authors also comment on the yield of their finally purified gp145, and how it compares to that of the SOSIP gp140 study (Ref 23)?

 3) Figure 6 is a native gel whose labeled banding is somewhat confusing. Inconsistent with the migration profile of the monomers observed by SDS-PAGE in Figure 3A, the observed monomers in Figure 6 seem to migrate close to the 480 kD marker. Why is there this discrepancy in the gels? Would it be possible to show the entire image of the native gel; ie. is there banding below the 242 kD marker that would be consistent with its monomeric weight?

 4) The sizing chromatography is mentioned in reference to oligomerization but the data are not shown. Can these data be shown beside the native gel in Figure 6? Also, in regards to the comment in Lines 480-482 how were the authors able to tell from the native gel that their gp145 has less aggregates compared to the reference standard? Is this interpretation based on a densitometry analysis of the higher order smear? Perhaps showing the SEC data alongside the native gel would help the reader draw a better comparison on this issue.

Minor comments:

- - Lines 613-614 and 615-616 are essentially duplicate sentences.

  - Provide a reference(s) for Lines 598-602.

  - Typos in Lines 111-112 and 168-169.
